# Recent Advances in Carbon-Based Iron Catalysts for Organic Synthesis

**DOI:** 10.3390/nano12193462

**Published:** 2022-10-03

**Authors:** Fei Wang, Fuying Zhu, Enxiang Ren, Guofu Zhu, Guo-Ping Lu, Yamei Lin

**Affiliations:** 1School of Food Science and Pharmaceutical Engineering, Nanjing Normal University, Nanjing 210023, China; 2School of Chemistry and Chemical Engineering, Nanjing University of Science and Technology, 200 Xiao Ling Wei Street, Nanjing 210094, China

**Keywords:** carbon material, iron, heterogeneous catalysis, organic synthesis, green chemistry

## Abstract

Carbon-based iron catalysts combining the advantages of iron and carbon material are efficient and sustainable catalysts for green organic synthesis. The present review summarizes the recent examples of carbon-based iron catalysts for organic reactions, including reduction, oxidation, tandem and other reactions. In addition, the introduction strategies of iron into carbon materials and the structure and activity relationship (SAR) between these catalysts and organic reactions are also highlighted. Moreover, the challenges and opportunities of organic synthesis over carbon-based iron catalysts have also been addressed. This review will stimulate more systematic and in-depth investigations on carbon-based iron catalysts for exploring sustainable organic chemistry.

## 1. Introduction

Organic synthesis has made great contributions to the progress of human society, but has also caused serious environmental pollution [1,2], so there is an urgent demand to introduce green chemistry into this area. One practical tool is the exploration of efficient and selective catalysts for organic synthesis, which can improve synthetic efficiency and reduce production costs and wastes, thereby minimizing the impact on the environment [3,4].

Iron, as the most productive and cheapest metal [5,6,7], is widely available, biocompatible and non-contaminant. The valence state of iron ranges from −2 to +5, which equips it with flexible redox properties, Lewis acidity and coordination ability [8]. Furthermore, iron-based enzymes are involved in lots of vital and efficient transformations *in vivo* [9]. All the above mentioned features of iron render a wonderful candidate to replace noble metals, and stand out among other non-precious metals. Therefore, iron-based catalysts meet the requirement of “catalyst economy”, and the development of these catalysts for organic transformations is conductive to the sustainable development of this area [6]. Nevertheless, the high chemical activity of iron also leads to its poor stability, which limits its application possibilities in more catalytic processes [10,11]. Meanwhile, iron has lower electronegativity and proportion of d orbitals and electrons than noble metals (such as Pd, Ru and Au), so it has poor catalytic activity in hydrogenation and cross-coupling reactions [12,13].

To solve these issues, ligands or supports are required to stabilize or tune the performance of iron sites [14,15,16,17]. Considering that ligands are relatively expensive and non-recyclable, solid supports are undoubtedly more in line with the requirements of green chemistry. Among various supports, carbon materials, with advantages of good biocompatibility, unique microstructure, high specific surface area and excellent physical and chemical properties, are considered excellent supports for iron-based heterogeneous catalysts [18,19,20,21,22,23,24,25]. Therefore, carbon-supported iron materials combining the advantages of iron and carbon materials are an efficient and sustainable catalyst for organic synthesis. In addition, *N*-doped carbon anchored iron-single-atom catalysts have similar FeN_4_ active sites with natural iron-based enzymes in terms of electronic, geometric and chemical structures [26,27].

Although there are several excellent reviews concerning the organic reactions over heterogenous and homogeneous iron-based catalysts [4,5,6,7,8,28,29,30,31,32,33,34,35,36,37], no attempt focuses on the carbon-based iron catalysts for organic synthesis. In recent years, there have been dozens of organic transformations catalyzed by carbon-based iron materials, including oxidation, reduction, tandem and other reactions. Taking into account the advantages and development trends of nanocarbon-based iron catalysts for organic synthesis, the review on this kind of heterogenous catalysts for organic reactions is of great importance and appeal.

In this review, we thus focus on the recent progress on nanocarbon-based iron catalyzed organic transformations, which is categorized by the reaction types. The introduction strategies of iron into carbon materials and the structure and activity relationship (SAR) are also highlighted in this review. In addition, the challenges and opportunities of organic synthesis over nanocarbon-based iron catalysts have also been addressed.

## 2. General Consideration

### 2.1. The Introduction Strategies of Iron into Carbon Materials

The catalytic activity of carbon-based iron materials is closely related to their structures, which are controlled by their synthetic strategies. Because numerous reviews on carbon-based metal materials have systematically summarized the controlled fabrication of these materials [38,39,40], this review mainly focuses on the introduction methods of iron into carbon materials. There are three main steps for the preparation of these catalysts: precursor synthesis, pyrolysis and post-modification (Figure 1). In the precursor synthesis step, iron can be introduced by mechanical mixing, self-assembly and impregnation methods [41,42,43]. Chemical vapor deposition (CVD) can be applied for iron doping during the pyrolysis process [44,45,46]. Impregnation is also an efficient approach for anchoring iron to carbon material in the post-modification step.

#### 2.1.1. Mechanical Mixing Strategy

Mechanical mixing refers to the direct mixing of iron sources and precursor compounds, without special treatment and requirements on the structure of the precursors. Although this method is easy to be operated, it results in the non-uniformity of carbon particle size and iron distribution [47]. Ball milling has been applied for more even mixing. In 2015, Deng et al. prepared FeN_4_/GN with highly dispersed FeN_4_ center by ball milling the mixture of iron phthalocyanine and graphene nanosheets (GNs) [48].

#### 2.1.2. Impregnation Strategy

The impregnation strategy is to impregnate the solution of iron source with carbon materials or carbon precursors, with the removal of residual solvent after impregnation equilibration then leading to the introduction of iron species into carbon materials or carbon precursors [49]. In general, the impregnation method has better mixing efficiency than mechanical mixing, resulting in a more uniform Fe doping. However, the impregnation method still cannot effectively solve the problem on the inhomogeneity of carbon particle size and iron distribution. For example, ethyl cellulose was dissolved in ethanol and mixed with the aqueous solution of Fe(NO_3_)_3_·9H_2_O, and then melamine and zinc nitrate were added as nitrogen source and pore former, respectively (Figure 2a) [50]. However, the size and distribution of carbon and iron particles in Fe@CN-Zn are not uniform, which can be observed from the results of TEM (Figure 2b) and SEM (Figure 2c).

#### 2.1.3. Self-Assembly Strategy

The introduction of iron can also be achieved through the self-assembly of iron salts with certain organic ligands to form ordered 3D porous metal-organic frameworks (MOFs), which can be divided into two types: Bimetallic MOFs and Fe-based MOFs. Compared to other methods, nanocarbon-based iron catalysts prepared from self-assembly precursors have advantages of ordered porous structure, more uniform carbon size and iron distribution [51,52,53]. However, this approach has special requirements for organic ligands, which should be able to coordinate with iron ions to form MOFs.

As a bimetallic MOF strategy example, Fe-ZIF (ZIF represents zeolitic imidazolate framework) was prepared via the self-assembly of iron salts and zinc nitrates with 2-methylimidazoles (Figure 3a) [43]. An *N*-doped carbon-based iron catalyst was synthesized by the pyrolysis of Fe-ZIF. As a representative example of Fe-based MOF strategy, MIL-101(Fe) was prepared by self-assembly of 1,4-terephthalic acid with iron nitrate hexahydrate, which was grinded with melamine and then calcinated to obtain Fe/Fe_2_O_3_@N^n^PC-T-x catalyst (n represents the ratio of MIL-101 (Fe) and melamine, T represents calcination temperature and x represents calcination pyrolysis time) (Figure 3b) [54].

#### 2.1.4. Chemical Vapor Deposition (CVD)

Chemical vapor deposition (CVD) refers to a vapor phase self-assembly carbonization process at high temperatures, which can effectively control the chemical composition, morphology, crystal structure and grain size of the membrane layer by adjusting the deposition parameters [55]. This method can make iron species in full contact with the carbon materials during the carbonization process, constructing a unique active iron center, and finally obtaining a catalyst for carbon-loaded iron with excellent catalytic activity. In 2021, Jia et al. prepared a Fe-N-C catalyst containing dense FeN_4_ sites by flowing iron vapor through the Zn-N-C substrate at 750 °C (Figure 4) [56].

### 2.2. The Structure and Activity Relationship (SAR)

#### 2.2.1. Nature Properties of the Carbon Materials

In comparison of metal oxide supports, carbon materials have unique porous structure, and are easier to be doped with heteroatoms, which can adjust their Lewis acid-base sites and defect the degree and electronic state of Fe sites, facilitating the formation of stable and active iron catalytic sites. Therefore, iron-doped carbon materials exhibit superior performance than metal oxide-supported iron catalysts in various transformations. For example, Beller’s group have prepared two carbon-supported iron catalysts: Fe-phenanthroline/C-800 [57] and Fe-L1@EGO-900 [41], which exhibit good catalytic activity for the reduction of nitroarenes and acceptorless dehydrogenation reactions, respectively. The two catalysts were prepared by calcinating the mixture of iron precursor, 1,10-phenanthroline ligand and carbon support. The difference is that carbon powder was used as carbon support in Fe-phenanthroline/C-800, while exfoliated graphene oxide (EGO) was applied as carbon support in Fe-L1@EGO-900. However, the catalytic efficiency of other heterogeneous iron catalysts is much lower, or even with no catalytic effect when the carbon material is replaced with other carriers, such as Al_2_O_3_, SiO_2_, CeO_2_ and TiO_2_. Overall, the porous structure and defects of carbon supports are favorable for the interaction between substrates with active sites, as well as the exposure of active sites in the catalysts. The Lewis acid-base sites of carbon supports can improve the adsorption and activation of substrates.

#### 2.2.2. Iron Sites in Carbon Materials

Generally, there are two types of iron sites in carbon materials: single atomic iron sites and iron nanoparticles (NPs) [58,59]. The identification of these two iron sites can be achieved by XRD, TEM, XAFS and HAADF-STEM [60,61,62]. Single atomic iron sites and iron nanoparticles (NPs) can usually play a synergistic catalytic role by exerting their respective catalytic advantages in different steps of one reaction [58,60,63]. Although some control experiments have been designed to verify the synergistic catalytic effect between these two sites, the research on this aspect is still in its infancy.

For example, Yang et al. designed a series of control experiment to confirm the synergistic catalysis on FeN_x_ sites (single atomic Fe sites) and Fe NPs for the fabrication of quinolones and quinazolinones (Figure 5a) [58]. Because acid etching can remove Fe NPs selectively, while SCN^−^ can specifically poison FeN_x_ sites, the catalytic effects of FeN_x_ and Fe NPs in the reaction can be clarified. In addition, the reaction principles of Fe(II)Pc and nano-Fe powder are similar to that of FeN_x_ sites and Fe NPs, which are also applied to detect the role of FeN_x_ and Fe NPs during the catalytic process. The coupling reaction of amine and aldehyde and the oxidative dehydrogenation of intermediate I are the two steps involved in the oxidative coupling reaction (Figure 5a). As shown in Figure 5a, FeN_x_ sites are more favorable for the formation of intermediate I, while Fe NPs are more conducive to the further dehydrogenation of intermediates I to produce quinolones and quinazolinones. Thus, FeN_x_ sites and Fe NPs play a synergistic catalytic role in this oxidative coupling reaction.

FeN_x_ site has similar structure with active sites of natural iron-based enzymes, so it exhibits excellent redox performance under mild conditions. Meanwhile, the coordination environment of iron single-atom sites can also be regulated by manipulating the preparation parameters, such as heteroatom doping, which may further improve the catalytic performance of iron sites. For example, Li et al. prepared phosphorus-doped atomically dispersed catalyst Fe-P-C for the first time, and found that the resulting O_2_-Fe-P_4_ structure is reduced by hydrogen to generate a large number of Fe-P_4_ sites, which is attributed to the excellent catalytic performance of the catalyst in the hydrogenation reaction (Figure 5b) [64]. However, the FeN_4_ sites generated in Fe-N-C prepared by nitrogen atom doping are less active towards the same reaction.

In general, FeN_x_ sites have higher atom utilization and catalytic efficiency than iron NPs, while Fe NPs in the catalysts do play a non-negligible catalytic role in many reactions. This is mainly because many organic reactions inherently involve multi-step steps that require the synergy of multiple active sites.

#### 2.2.3. The Construction of SAR

The construction of SAR between carbon-based iron catalysts and organic reactions are mainly explored by control experiments, reaction kinetics studies and density functional theory (DFT). Control experiments are performed to determine possible reaction intermediates and reaction pathways by changing the reaction conditions or substrates appropriately, or by adding additives. As shown in Figure 1, the role of Fe-Fe_3_C@NC-800 and H_2_O_2_ in the reaction is investigated by control experiments. Based on the results of Figure 1, it can be concluded that Fe-Fe_3_C@NC-800 is necessary for both the coupling and dehydrogenation process, while H_2_O_2_ is favorable for the dehydrogenation process [58].

The main purpose of reaction kinetic studies is to determine the rate-determining step (RDS) of the reaction. RDS is the most critical step in the complex organic reactions, so determining RDS is beneficial to simplify the studies of the reaction mechanism. Kinetic isotope experiments (KIE) is a commonly used approach to determine RDS, in which the k*_H_*/k*_D_* value can be an indicator for RDS [65]. The RDS can also be determined by detecting reaction orders, which reflects the effect of reactant concentration on the reaction rate. For example, our group determines the borrowing hydrogen *N*-alkylation reaction order of benzyl alcohol and benzylamine, and performs KIE to confirm the RDS (Figure 6) [66]. The reaction order of aniline in this reaction is negative, while that of benzyl alcohol is positive (Figure 6b,c), confirming the activation of benzyl alcohol may be involved in the RDS [67]. The parallel experiments are performed by using PhCH_2_OH and isotope-labeled PhCD_2_OH to go through this reaction respectively, and the k*_H_*/k*_D_* of this reaction is calculated to be 2.75 (Figure 6a), further indicating that the C-H activation of benzyl alcohol may be related to RDS.

In addition, DFT calculations can further confirm the main catalytic active center in RDS by comparing the energy barriers, which can also provide useful information for the construction of SAR between carbon-based iron catalysts and organic reactions. For instance, to further identify the catalytically active center, the RDS energy barriers of different Fe sites can be calculated based on DFT [4,43,68,69,70]. The energy barriers of these four sites can be sorted as Fe_1_-N_4_S_1_ < Fe_1_-N_5_ < Fe_1_-N_4_O_1_ < Fe_2_O_3_, which are positively related to their positive charge density (Figure 6d,e). These results suggest that Fe_1_-N_4_S_1_ and Fe_1_-N_5_ are more likely to be the main catalytically active sites for the RDS process [66].

Based on these results, a possible strategy for construction the structure-activity relationship is listed: (1) confirming the rate-determining steps of reactions and the main active sites of carbon-based iron catalysts by catalyst characterization data, control and kinetic experiments; and (2) establishing the relationship between the two based on reaction yields and structural information of the active sites, and confirming it by DFT calculations.

## 3. Oxidation Reactions

Compared to metal oxides, O_2_ and metal-free peroxides are undoubtedly greener and safer oxidants, but their own oxidative capacity is relatively poor. Thus, further activation of these oxidants by certain catalysts is required. Iron features the ability to transfer electrons to O_2_ and peroxides, thereby displaying excellent performance on many oxidation reactions. Furthermore, the porous carbon materials are favorable for the adsorption of O_2_ and peroxides. Therefore, carbon-based iron catalysts show superior catalytic performance than noble metal-based catalysts in many oxidation reactions using O_2_ or metal-free peroxides as oxidizing agents.

### 3.1. Selective Oxidation of Nitrogenous Compounds

In 2015, FeO_x_ NPs surrounded by nitrogen-doped-graphene shells that immobilized on carbon layers (FeO_x_@NGr-C) were successfully prepared by simple pyrolysis process (Figure 2a). This catalyst was applied in the oxidative dehydrogenation of *N*-heterocycles, which overcomes the poor functional-group tolerance and harsh condition requirements involved in the reported catalytic system (Figure 2b) [17]. The high thermal stability and coordination ability of 1,10-phenanthroline promotes the formation of graphene shells around FeO_x_ NPs during the pyrolysis process, and the formed *N*-doped carbon layers with graphene-stacking defects lead to the exposure of active sites for O_2_ absorption. In the same year, this group developed a similar catalyst (Fe_2_O_3_/NGr@C) to achieve the selective dehydrogenation of primary amines for the production of various aliphatic, aromatic and heterocyclic nitriles (Figure 2c) [71]. It is worth noting that the use of excess aqueous ammonia can inhibit the formation of secondary imine, thereby improving the reaction selectivity. In addition, the unique carbon layer encapsulation structure makes the activity of Fe_2_O_3_/NGr@C not significantly decrease after 5 runs, exhibiting industrial application potential.

### 3.2. Epoxidation of Olefines

The epoxidation of olefines provides a variety of important building blocks for the pharmaceutical and fine chemical industries [72]. In 2019, Kucernak et al. introduced a Fe-N/C catalyst by calcining the complex of polymerized 1,5-diaminonaphthalene and FeCl_2_ [73]. The FeN_x_ active sites in Fe-N/C are similar to that in Fe porphyrins, and thereby display excellent performance in the epoxidation of olefines, with O_2_ as the oxidant under mild conditions (Figure 7a). This is also the first example that such kind of catalysts are used for the aerobic epoxidation of olefins under mild conditions, which with high turnover frequency (2700 h^−1^) and good recyclability can be reused 5 times.

An iron single-atom site catalyst (SAS-Fe) with 30 wt.% of Fe loading was fabricated by the reductive calcination of the complex formed by the coordination of dicyandiamide and Fe(NO_3_)_3_·9H_2_O (Figure 7b) [74], which overcomes the shortcomings of common SAS catalysts, such as low metal loading, and the aggregation and movement of single-atom-sites. Furthermore, DFT calculations were performed to study the catalytic mechanism of SAS-Fe (Figure 8). It was found that the difference in the energy barrier to generate epoxy styrene or benzaldehyde during the reaction can reach a maximum of ΔE_a_ = 0.83 eV, revealing that SAS-Fe tends to promote the formation of epoxy styrene rather than phenylacetaldehyde. The selectivity of SAS-Fe to epoxy styrene can be as high as about 90%, and its activity remains unchanged after 5 cycles. Due to the high metal loading and scalable production, SAS-Fe does meet the requirement of industrial applications.

### 3.3. Selective Oxidation of Hydrocarbons

In 2015, Deng et al. fabricated highly dispersed single FeN_4_ centers by high-energy ball milling of the mixture of iron phthalocyanine (FePc) and graphene nanosheets (GNs) under controlled conditions. The obtained catalyst FeN_4_/GN exhibits excellent activity for oxidizing benzene to form phenol at room temperature using H_2_O_2_ as oxidant [48]. The catalytic mechanism of FeN_4_/GN is proposed based on the results of control experiments and DFT calculations, indicating that the formation of Fe=O moiety is a critical step to realize the transformation of benzene to phenol (Figure 9a).

Nanocarbon-supported iron catalysts were found to show “platinum-like” properties in catalytic reactions, while the specific active site on them still remains controversial. Along this line, Zhang et al. synthesized an atomically dispersed Fe-N-C catalyst for the selective oxidation of C-H bond in alkanes with high selectivity [60]. In order to investigate the main active site in Fe-N-C, a series of characterizations are performed, the results of which indicate that the atomic iron mainly exists in the form of FeN_x_ (x = 4–6) (Figure 9b). Based on control experiments, FeN_5_ shows the highest TOF compared to FeN_4_ and FeN_6_, although the content of FeN_5_ is much lower. In addition, the content of FeN_5_ sites decreases as the temperature raises from 700 °C to 800 °C, which in turn results in a significant loss of catalytic activity. It can be concluded that the content of FeN_5_ depends on the calcination temperature, and can significantly affect the catalytic efficiency.

## 4. Reduction Reactions

In addition to oxidation reactions, nanocarbon-supported iron catalysts also show potential to catalyze reduction reactions [75,76]. Hydrodeoxygenation, hydrogenolysis and hydrogenation are three main kinds of catalytic reduction reaction over iron-doped carbon materials.

### 4.1. Hydrodeoxygenation

In 2017, Fu et al. described an effective iron catalyst (Fe-L1/C-800) by simultaneous calcination of 1,10-phenanthroline (L1) as the nitrogen precursor and activated carbon as the carrier at 800 °C, which could selectively realize the hydrodeoxygenation of HMF (5-hydroxymethylfurfural) to DMF (*N, N*-dimethylformamide), with a selectivity of 86.2% [77]. The hydrogenation of aldehyde group in HMF is found to be the rate-determining step, which can be accelerated by the use of alcohol as an additional hydrogen source. Moreover, this catalyst maintains excellent stability and high selectivity in both batch and continuous flow fixed-bed reactors.

The selective hydrodeoxygenation (HDO) of carboxylic acids to alkanes is of great importance for the production of biofuel from biomass [78]. Therefore, a *N*-doped carbon-alumina hybrid supported iron (Fe-N-C@Al_2_O_3_) catalyst was prepared by calcining the mixture of Fe(acac)_3_, melamine and Al_2_O_3_ at 900 °C [79]. Surprisingly, the Fe-NC@Al_2_O_3_ catalyzed HDO of carboxylic acids displays excellent chemo-selectivity, even in the presence of an aromatic ring that is generally easier to be hydrogenated than carboxylic acid (Figure 10), which is ascribed to the formation of the FeC_3_ active phase and *N*-doped carbon-alumina hybrid. Moreover, the formation of both nitrogen-doped carbo-alumina hybrid and iron loading effect the Lewis basicity of Fe-N-C@Al_2_O_3_ to adsorb substrates.

### 4.2. Hydrogenation of Nitroarenes

Beller et al. developed a classic iron catalyst (Fe-phen/C) derived from pyrolyzing the mixture of Fe(OAc)_2_, 1,10-phenanthroline and carbon powder (Figure 11) [42]. The prepared catalyst exhibits high chemo-selectivity toward the reduction of various nitroarenes to corresponding aromatic amines with hydrazine hydrate as a hydrogen source. Subsequently, the catalytic activity toward the reduction of nitroarenes of Fe-phen/C was further investigated by the same group using H_2_ or HCOOH as a hydrogen source [57,80]. The FeN_4_ sites on the iron oxide surface are the source of the unique catalytic activity of this catalyst. Meanwhile, iron oxide can also play a minor role in the reduction reaction if it does not inhibit the FeN_4_ sites of the catalyst.

It is well known that nitrogen atoms act as basic sites in *N*-doped nanocarbon-supported iron catalysts, which can improve their hydrogenation performance [81]. Melamine was mixed with MIL-101(Fe) to improve the nitrogen content, and the resultant mixture was pyrolyzed at 700 °C to obtain Fe/Fe_2_O_3_@N^5^PC-700-1 [54]. This material exhibits excellent chemical selectivity towards the hydrogenation of nitro compounds, and the turnover frequency (TOF) reaches 8898 h^−1^, which is 100 times higher than that of reported similar catalysts [82]. Meanwhile, the nitrogen sites on the catalyst surface can effectively capture nitrobenzene and aniline to further improve the catalytic activity of this material.

In 2019, a novel iron single-atom catalyst Fe_1_/N-C was prepared by immersing the template SBA-15 into the mixture of ferric nitrate and glucosamine, which was then calcinated at inert atmosphere before acid etching [70]. The prepared Fe_1_/N-C exhibits excellent activity for the transfer hydrogenation of nitroarenes to amines with hydrazine as a hydrogen source. A series of characterization confirmed that the formation of FeN_4_ active sites in Fe_1_/N-C, and the pathway for hydrogenation of nitroarene in the presence of FeN_4_ sites was investigated [83].

In 2020, Fe_SAs_/Fe_2_O_3ACs_/NPC that both contain FeN_4_ sites and Fe_2_O_3_ clusters was fabricated by the pyrolysis of Fe-ZIF at 900 °C, which was then annealed naturally [43]. Compared with Fe_SAs_ and Fe NPs, Fe_SAs_/Fe_2_O_3ACs_/NPC shows the highest activity towards hydrogenation of nitroarenes (TOF up to 1923 h^−1^), owing to the synergistic effect between Fe_SAs_ and Fe_2_O_3ACs_. Based on the DFT results, Fe_SAs_ reduces the overall barrier for hydrazine to decompose into hydrogen, and then under the catalysis of Fe_2_O_3ACs_, the hydrogen further participates in the hydrogenation reaction of nitrobenzene.

In 2022, our group developed an aniline modified ZIF-derived *N*-doped carbon iron single-atom catalyst (Fe_SA_@NC-20A) with 2.4 wt.% Fe loading (Figure 12) [84]. This catalyst displays excellent catalytic performance on the selective hydrogenation of nitroarenes, with N_2_H_4_·H_2_O as the reducing agent (TOF up to 1727 h^−1^, more than 10 runs). This strategy also exhibits great potential to be applied to the synthesis of complex amines and drugs. According to DFT calculations and control experiments, the N-H activation of N_2_H_4_·H_2_O is confirmed as the rate determine step, and FeN_4_ sites in Fe_SA_@NC-20A are considered the active sites for this reaction.

Recently, a mesoporous carbon (MC) support was prepared by hydrothermal reaction of citric acid and magnesium citrate, which was then loaded by potassium ferrocyanide through calcination to form NMC-Fe [85]. The prepared NMC-Fe could selectively mediate the reduction of nitro aromatics into azo compounds using hydrazine hydrate as a hydrogen source (Figure 3), which was due to the formation of Fe-N complex in the NMC-Fe catalyst.

### 4.3. Others

In 2016, the group of Fu applied an iron catalyst (Fe-L1/C-800, the same catalyst in ref. [77]) to achieve the transfer hydrogenation of furfural (FF) to furfuryl alcohol (FFA) with a selectivity of 83.0% and a conversion rate of 91.6% [86]. Moreover, the catalyst can be reused five times, and finally deactivates due to the destruction of the Fe-N bond, the formation of crystalline Fe_2_O_3_ phase and the change of pore structure. In 2018, they continued to study the same catalyst for the selective cracking of C-O bonds in lignin model compounds (Figure 4). In the presence of Fe-L1/C-800, the α-O-4 linkage of lignin model compounds is directly hydrolyzed to phenol and toluene, with yields of 95% and 90%, respectively [68]. In the occasion of the β-O-4 compounds, the presence of vicinal -OH group is required. Although monomers are not detected during the Fe-L1/C-800-catalyzed lignin pyrolysis process, its selectivity to aromatic hydrocarbons is still much higher than that of noble metal catalysts.

## 5. Cascade Reactions

Cascade reaction is considered as a sustainable and environmentally friendly tool for the synthesis of new drugs and natural products, owing to its step and atom economy, simple operation process, lesser amount of solvent and reagents used and minimized waste emission. Therefore, the preparation of secondary amines, nitriles, unnatural amino acids and *N*-arylsulfonamides has been explored via a cascade reaction catalyzed by nanocarbon-supported iron catalysts.

### 5.1. Dehydration-Oxidation Reactions

Li et al. prepared 2,5-diformylfuran (DFF) from fructose via one-pot two-step process with 5-hydroxymethylfurfural (HMF) as the intermediate (Figure 13a) [87]. This transformation could be achieved by an octahedral Fe/C-S catalyst, which was produced from the pyrolysis of sulfur powder-doped Fe-based MOF (Figure 13b). The doping of sulfur results in more acidic sites that promote the dispersion of iron species, which is beneficial for the dehydration of fructose and the oxidation of HMF to DFF. The high catalytic activity and selectivity for Fe/C-S were also attributed to the strong adsorption of HMF and weak interaction of DFF with catalyst, respectively. Furthermore, the strategy for catalyst preparation applied in this work provides new insights to synthesize uniform-shaped metal nanoparticle catalysts from MOFs.

### 5.2. N-Alkylation Reactions

Our group prepared a stable and recoverable N, S-codoped Fe_20_-SA@NSC catalyst with a high iron loading of 2.51 wt.% [66], which can catalyze the *N*-alkylation of amines and alcohols with high selectivity and efficiency under solvent free conditions (TOF up to 13.9 h^−1^) (Figure 14). According to the results of HAADF-STEM, XAFS and DFT calculations, the single atom site Fe_1_-N_4_S_1_ was formed in Fe_20_-SA@NSC with high electron density, which not only enhances the C-H bond activation of alcohols, but also improves the hydrogen borrowing ability.

In 2014, Beller et al. synthesized Fe_2_O_3_/NGr@C by their reported method for the *N*-alkylation of benzaldehyde with nitroarenes to prepare secondary amines [88]. In this work, anilines were formed in situ from the hydrogenation of nitroarenes, and underwent a condensation reaction with aldehydes to afford imines, which were then reduced to afford *N*-alkylation amines (Figure 5a). This method shows atom economy and environmental friendliness, providing an important application prospect for the industrial production for secondary amines. In 2017, they continued to study the application of Fe_2_O_3_/NGr@C catalyst in the fabrication of *N*-methylated amines from nitroarene and paraformaldehyde without a hydrogen source [89]. The paraformaldehyde in this work acts as both a hydrogen source and a methylating agent. In addition to structurally different substrates, some drugs and fluorescent molecules containing nitro groups can also well undergo this strategy to afford corresponding *N*-methylated amines (Figure 5b), including nimodipine, cilnidipine, nicardipine, nimesulide, rhodamine derivatives and fluorenone.

### 5.3. Oxidative Coupling Reactions

In 2018, the nitriles and amides were prepared via the tandem reaction of aldehydes and ammonia in the presence of Fe_2_O_3_-N/C [90]. The aldehydes and ammonia first undergo condensation reaction to generate imines, which are oxidized to desired nitriles (Figure 6a). An additional hydrolysis process is required for the formation of amides (Figure 6b). Both the nitrogen-doped graphene layer-supported Fe_2_O_3_ nanoparticles and Fe-N interactions are found to contribute to the activity of Fe_2_O_3_-N/C. Moreover, gram scale synthesis is realized in this green strategy, and the Fe_2_O_3_-N/C can be reused for several times, indicating the potential industrial application of this work.

In 2022, some of us synthesized a single-atom catalyst Fe_1_-N-C by calcinating ZIFs to achieve the conversion of a wide range of primary alcohols to nitriles under mild conditions for the first time (Figure 15) [91]. In the preparation process of the catalyst, benzyl amine is introduced to make the obtained catalysts with higher nitrogen contents, and modify the morphology, crystal structure and size of ZIFs, which is conducive to the formation of stable FeN_4_ active sites. The catalyst facilitates the absorption of oxygen and the transportation of reactants in the catalytic process, and maintains a high catalytic activity after being reused six times, which plays an important role in exploring the conversion of renewable biomass resources (bio-based primary alcohols) into fine chemicals (nitriles).

In 2019, Fe-Fe_3_C@NC, composed of a *N*-doped carbon layer surrounded Fe-Fe_3_C nanoparticles and FeN_x_ sites, was prepared by Song et al., which was then applied in the oxidative cyclization reaction between amines and aldehydes to efficiently synthesize quinolines and quinazolinones [58]. A series of characterization and control experiments were performed to investigate the catalytic mechanism, the results of which indicate that the Fe-Fe_3_C nanoparticles and FeN_x_ sites in Fe-Fe_3_C@NC have a synergistic catalytic effect in catalyzing this cyclization reaction. This strategy provides a convenient and sustainable method for obtaining a set of *N*-heterocycles with pharmaceutical action.

Natural woods are an abundant biomass with hierarchical structure and good mechanical properties, which have been applied as porous carbon supports to immobilize iron single atoms for the formation of iron-based single atom catalyst (SAC) (Figure 16) [92]. The obtained iron-based SAC shows excellent performance in three-component oxidative cyclization to afford various quinolines from anilines and ketones under mild conditions. In addition, this iron-based SAC exhibits better activity than traditional catalysts, such as Pd/C, Fe/C and FeCl_3_. The DFT calculations were performed to identify the active sites in iron-based SAC, and the FeN_4_ sites were found with lower reaction barriers than Fe-C_1_N_3_ and Fe-C_2_N_2_. Therefore, the excellent catalytic activity of iron-based SAC can be attributed to the weak desorption energy between FeN_4_ and products, and the increased exposure of active sites resulted from hierarchically porous carbon plates.

## 6. Others

Acceptorless dehydrogenation can produce valuable intermediates while generating molecular hydrogen, which is known as a green energy resource. Based on this, Balaraman et al. prepared a Fe-L_1_@EGO-900 catalyst with iron oxides as the outer shell and iron carbide as the inner core for acceptorless dehydrogenation of *N*-heterocycles, amines and alcohols (Figure 7) [41]. However, other iron catalysts prepared from traditional supports, such as Al_2_O_3_, SiO_2_, CeO_2_ and TiO_2_, failed to undergo this reaction under similar experimental conditions. This is due to the fact that 1,10-phenanthroline (L_1_) with broad *π* area is applied to prepare Fe-L_1_@EGO-900, which enables the catalyst to better recognize and bind substrate. Furthermore, this catalyst can be reused four times without a significant decrease in activity, showing potential for practical application.

In 2022, Beller et al. prepared Fe-Cellulose-1000 with metallic iron as the active site by pyrolyzing the mixture of cellulose and Fe(NO_3_)_3_·9H_2_O at 1000 °C [93]. This material breaks through the limitations of the acid-required hydrogen-deuterium exchange reaction, and can selectively catalyze the deuteration of the locations that are not prone to electrophilic substitution reactions (Figure 17). In the reaction pathway of catalytic deuteration, the iron oxides in Fe-Cellulose-1000 can be reduced in situ to metallic iron under hydrogen pressure, which further promotes the homolytic splitting of the deuteration reagent (D_2_O). The generated •OD radicals are activated and adsorbed on the surface of the catalyst in the form of D* and *OD, and the former one can selectively substitute hydrogen atom on the benzene ring. Therefore, the Fe-Cellulose-1000 can promote highly selective deuteration reaction of aniline, indole, phenol and other heterocyclic compounds, and there is no obvious activity loss of Fe-Cellulose-1000 after five times of recycle. Furthermore, this catalyst can be applied to the preparation of kilogram-grade deuterated compounds, providing the possibility of industrial production.

Another efficient catalyst Fe@NC-800 for acylation reaction was reported by Jia et al., which was prepared by pyrolyzing the Fe-DABCO-MOF formed by self-assembly of FeCl_2_·4H_2_O, terephthalic acid and 1,4-diazabicyclo [2,2,2] octane (DABCO) at 800 °C (Figure 18) [94]. In addition to provide nitrogen source, the incorporation of DABCO can also facilitate the formation of highly graphitized materials and inhibit iron agglomeration. The excellent catalytic performance of this catalyst is mainly attributed to the electronic effect of the graphic layer and encapsulation of iron particles by the graphic layer, which will promote the stabilization of the iron species.

In 2021, our group constructed a nitrogen-doped carbon supported iron-based catalyst Fe@CN-Zn using iron nitrate as the iron source, zinc nitrate as the pore forming agent, and melamine and ethyl cellulose as the nitrogen source and carbon source, respectively [50]. In this work, carbene species were generated from diazoacetate in the presence of Fe@CN-Zn, and then further reacted with amines to afford corresponding amino acid derivatives (Figure 19). The *N*-doped carbon decreased the electron density of Fe/FeO_x_ nanoparticles in Fe@CN-Zn [95], which is beneficial for the formation of intermediate A. Meanwhile, the dope of nitrogen afforded more Lewis base sites, improving the transfer of proton during the rection process. The active Fe@CN-Zn catalyst can be recycled four times without significant activity loss.

## 7. Summary and Outlook

This review focuses on organic reactions catalyzed by carbon-based iron materials, such as oxidation, reduction and tandem reactions, in which the introduction strategies of iron into carbon materials and the structure and activity relationship between carbon-based iron catalysts and organic reactions are summarized. Despite the great development in this field, challenges and opportunities for the widespread application of iron-doped carbon materials still exist.

The homogeneity and tunability of iron sites are the basis for the high catalytic performance of carbon-based iron catalysts. There is still a lack of effective means to regulate and determine the coordination environment of iron sites. A feasible solution is anchoring single iron atoms on nitrogen-doped carbon, forming FeN_x_ sites with similar coordination environments, which can be detected by X-ray absorption near edge structure (XANES). Nevertheless, limited approaches have been demonstrated for fine-tuning FeN_x_ sites. At the same time, the current characterization of XANES is the weighted average of the coordination environment of all iron sites, so the homogeneity of iron sites is actually not high.

Compared with homogeneous iron catalysis, the application range of carbon-supported iron catalysts in organic synthesis is very limited. There are two main reasons for this:

(1) The poor homogeneity and tunability of iron sites in these catalysts limits its application in complex and delicate organic synthesis, such as asymmetric synthesis. The investigation of efficient strategies for improving the homogeneity or tuning the coordination environment of iron sites is desirable.

(2) Since most organic chemists are not proficient in designing and preparing such materials, more chemists that are specialized in organic synthesis, catalysis and materials ought to undertake related cooperation and innovation in this field, in order to expand its application range and depth.

Finally, carbon-based iron catalysts also offer more new possibilities for organic synthesis. For example, these catalysts exhibit excellent catalytic activity for the activation of O_2_, owing to porous structure and enzyme-like iron sites, so they can enable many efficient green oxidation processes.

## Data Availability

All data generated or analyzed during this study are available within the article.

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
