# Peer review of "Recent Advances in Carbon-Based Iron Catalysts for Organic Synthesis"

_nanomaterials, 2022, doi:10.3390/nano12193462_

Round 1

Reviewer 1 Report

The review calls 'Recent advances in carbon-based iron catalysts for organic synthesis" is an inventory of the procedure used for producing some iron carbon heterogeneous catalysts, and their use in organic synthesis. 

The article is well written, there is a good mix between pure catalysis subject and organic synthesis subject. 

The review will earn more visibility by adding color in the organic synthesis scheme (color for the group which undergo reaction etc...)

The font is not uniform in all the review, please correct. The size of the molecules is not the same, please correct this. 

With these small change, the article can be published in nanomaterials.

Reviewer 2 Report

The authors review recent works on application of heterogeneous iron catalysts on carbon-based support in organic synthesis. The idea is interesting, but the manuscript needs revision to make this review informative for the reader.

- Please correct languages mistakes, they prevent the reader from following the authors' ideas. Some examples of sentences which need revision:

Abstract, lines 17-20: "The present review ... highlights the introduction strategies of iron into carbon materials..."

Lines 108-110: "Iron salts can self-assemble with certain organic ligands to form ordered 3D porous metal-organic frameworks (MOFs), which can be divided into two ways: Bimetallic MOF strategy and Fe-based MOF strategy."

Lines 216-218: "For instance, the different Fe sites resulted energy barriers of RDS were calculated based on DFT to further define the catalytic active center [4,43,68-70]."

Lines 224-225: "Thus, the further activation of these oxidants by the catalysts are norm."

Lines 232-234: "In 2015, nitrogen-doped graphene shells surrounded FeOx NPs that immobilized on carbon (FeOx@NGr-C) was successfully prepared by simple mixing and pyrolysis processes (Scheme 2a)."

Lines 238-240: "and the resulted N-doped carbon layers contains graphene stacking defects, leading to the emergent of active sites for O2 absorption."

Lines 333-334: "Meanwhile, iron oxides can also play a minor role in the reduction reaction if they didn’t inhibit the FeN4 sites of the catalysts."

Lines 424-426: "The doping of sulfur resulted in more acidic sites to promote the dispersion of iron species and the dehydration of fructose..."

There are also many other fragments which need revision in terms of language and style, please check carefully throughout the manuscript.

- The passage in lines 400-409 reviews works on SiO2-supported Fe-based catalysts. This seems to be outside the scope of the review, which is devoted to carbon-based materials as the support for Fe-based catalysts.

- In several examples (lines 154-168, 362-364, 407-409, 491-492), the authors postulate "a synergistic role" of iron species of different types in the catalysis. Please provide more clear description of this synergy.

- The authors attempt to analyze structure-activity relationship for the catalytic activity of iron-based catalysts on carbon-based supports. Please provide more clear conclusion from this analysis. What factors in chemical structure of the catalyst affect the relevant parameters of the catalyst activity?

Reviewer 3 Report

The manuscript reviews recently reported strategies for the use of carbon-based iron catalysts in diverse organic synthesis fields. On this objective, it is timely, well-organized and fulfils the general requirements for a review on the subjected theme. The paper is particularly useful on the description of methods for preparing such materials, in conjunction with selected applications, giving plenty and appropriate literature references to those interested in pursuing such subject, and highlighting the several limitations of each method. The paper will be of particular interest to organic chemists driven by sustainable/green approaches to a series of important organic chemistry transformations, wishfully working in cooperation with material science scientists, as outlined in the Outlook. We recommend the acceptance of the manuscript, pending minor corrections on the English, text formatting (which should be carefully revised), and typos. Besides, some Figures need to be properly revised; for instance, the schemes appearing in Fig 7 (c) and Fig. 11 for the sake of legibility.

Round 2

Reviewer 2 Report

The authors have addressed my concerns.

Minor remarks:

1. Please check the manuscript for typos, some examples:

- line 58: "none" -> "no"

- line 61: "accounts of" ->"account"

- line 382: "highly" -> "high"

2. Please add more information on the notations of the catalysts. For example, what is the meaning of "1000" in "Fe-Cellulose-1000" catalyst name (line 572)? Or what is the meaning of "800" in "Fe@NC-800" (line 591)? Does "Zn" stand for "zinc" in "Fe@CN-Zn" (line 603)? If so, where does this zinc comes from, it is not clear from the description given in lines 603-610.

3. Please expand the "ZIF" notation in line 121, where it appears for the first time.

4. Please explain the composition of Fe-L1@EGO-900 when this notation appears for the first time (line 155). At the moment, the explanation is given only in lines 561-562.
